# Elementary School Teachers' Self-Assessment of Use of Positive Behavior Support Strategies and Goal Setting Related to Equity-Focused Features

**Julie Sarno Owens** [1,*] , **Deinera Exner-Cortens** [2] , **Madeline DeShazer** [1] , **John Seipp** [1] , **Elise Cappella** [3] , **Natalie May** [3] and **Nick Zieg** [1]

1   Psychology Department, Ohio University, Athens, OH 45701, USA; md956919@ohio.edu (M.D.); seippj@ohio.edu (J.S.); nz076815@ohio.edu (N.Z.)
2   Department of Psychology, University of Calgary, Calgary, AB T2N 1N4, Canada; deinera.exner2@ucalgary.ca
3   Department of Applied Psychology, Steinhardt School of Culture, Education, and Human Development, New York University, New York, NY 10003, USA; elise.cappella@nyu.edu (E.C.); natalie.may@nyu.edu (N.M.)
*   Correspondence: owensj@ohio.edu; Tel.: +740-593-1074

**Abstract:** The goal of the Maximize Program is to collaborate with educators to develop resources and procedures to facilitate teachers' use of equity-focused behavioral supports. In this study, we describe teachers' responses to the first iteration of the interactive Maximize Technology Platform. Ninety elementary school teachers from three schools were encouraged to use the platform to learn about the foundational concept of equity literacy, complete a self-assessment of practices, and set a goal for improvement. We observed teachers' platform use, self-reported use of 10 behavior support strategies, goals set for improving equity-focused features of these strategies, and reported progress during the first quarter of the academic year. Over 70% of teachers reported frequent use of four strategies: Classroom Expectations, Praise, Greetings, and Community Circles. Fewer teachers reported using Student Choice, Effective Questioning, and Corrective Feedback. Variations in use between general education and other teachers were observed. Over 60% of teachers set an equity-focused goal. Variability in the types of goals set and rates of reported improvement highlight the complexity of this work. Results offer promise about the use of interactive technology to facilitate professional learning and goal-setting about equity initiatives and offer insights for leveraging interactive technology to facilitate teachers' implementation of equity-focused practices.

**Keywords:** positive behavior supports; equity; teachers; self-assessment; classroom management; technology

## 1. Introduction

### 1.1. Inequity in Educational Experiences and Outcomes

Educational equity means that student success or failure is not connected to any social or cultural factor (e.g., gender, race, ethnicity, religion, sexual orientation, language, or family economic status) [1]. It also means that all students have equal access to an extensive range of learning opportunities and materials, with fair distribution of these resources based on need and fair processes for distributing resources, such that students receive equal services for equal need (i.e., horizontal equity) and enhanced services for greater need (i.e., vertical equity) [2]. From an experiential perspective, educational equity means that all students feel seen, heard, welcomed, connected, included, engaged, and valued in their school each day.

Given the history of the United States and the impact that colonialism, patriarchal values, and white supremacy have had on the development of our social systems [3–6], it is not surprising that students' current educational experiences and outcomes remain

connected to sociocultural factors. For example, Black and Latino boys are nearly four times more likely than white and Asian students to be expelled from school, and Black girls are suspended at higher rates than girls of any other race or ethnicity [7–9]. Children living in poverty have significantly lower school performance at all grade levels than their financially resourced counterparts [10,11]. Achievement disparities worsened after the pandemic, with students in economically and racially minoritized communities left even further behind in math and reading than fellow students in wealthier, whiter districts [12]. In addition, access to resources is also connected to sociocultural factors. For example, children living in economically marginalized communities have less access to libraries and related educational supports as compared to children living in middle- to upper-income communities [10,13]. Given these statistics and the lived experiences of students who experience inequity on a daily basis in schools [14,15], we believe that continued work toward creating more equitable educational experiences is needed.

*1.2. Change Efforts*

The serious challenges experienced by students in our nation's schools are complex and require change at all levels (federal, state, and local). Although achieving equity requires systemic change and the dismantling of systems of oppression [16], because societal change is slow and often comes with setbacks after successes, we also need to determine what we can do to spark change at the individual level, one teacher at a time. Namely, regardless of the facilitators or barriers at broader levels (e.g., access to materials, resources, and services), teachers have control over how they interact with their students and which instructional and classroom management strategies they use. Ample research shows that several teacher-controlled aspects of the educational experience (e.g., positive behavior supports, opportunities for engagement) have the potential to improve students' academic achievement and positive experiences in the classroom [17–20]. As such, teacher autonomy represents a critical opportunity for professional learning and growth for teachers and a leverage point for implementing equity-focused strategies to improve educational experiences and outcomes for students.

The goal of the Maximize Program is to collaborate with educators to develop resources and procedures to help teachers maximize their use of equity-focused behavior supports. Specifically, we are collaborating with a multi-region advisory board with members representing diverse lived experiences and student populations and with extensive experience in equity-focused change efforts in schools. We are building on strengths that lie within the positive behavior supports and intervention (PBIS) movement and encouraging teachers to reflect on how these strategies can be modified to enhance students' experiences of feeling welcomed, included, and valued and to reduce educational inequity. We are conducting this work in the context of Central Ohio, where the teaching staff is over 80% white women and the student bodies are diverse in race, ethnicity, economic, and language backgrounds. Staff at these schools have experienced some professional development on diversity and inclusion, and some have participated in additional efforts toward equity-focused change (e.g., professional learning communities, book studies). However, all principals in the project reported that receptivity to these efforts is mixed and that meaningful change has been slow. We kept this in mind while developing the platform, giving considerable thought to how to enhance receptivity among teachers who did and did not have experience with equity-focused work.

Within this context, we were interested in the extent to which an interactive technology-based platform (The Maximize Platform) could provide (1) a private space to facilitate teachers' professional learning about a foundational equity literacy framework (see www.equityliteracy.org) and self-reflection and (2) tools to encourage self-assessment of practices, motivation for goal setting, and progress monitoring to simulate meaningful changes in teacher practices. Given the important role of self-reflection in teacher professional development [21,22] as well as evidence that "required" trainings on diversity-related topics and shame-inducing efforts can have unintended consequences [23,24], we priori-

tized teacher autonomy with regard to their interaction with the platform's features and activities. In addition, given the reported mixed receptivity to equity-focused initiatives, we titrated the professional learning content by placing familiar content (i.e., self-assessment of traditional positive behavioral support strategies) early in the user journey with more thought-provoking content (i.e., key features for equity) later in the user journey (i.e., on the "Learn More" pages).

The goal of the current study is to examine how teachers interacted with the initial version of the Maximize Platform, their self-reported use of 10 positive behavior support strategies, and their interest and willingness to set a goal for improving an equity-focused feature of a given strategy during the first quarter of the 2022–2023 school year. This paper represents the first in a line of studies examining the utility of multiple strategies and resources designed to move the needle on teachers' use of equity-focused positive behavior support strategies. Lessons learned from each study will inform the development of a larger collection of change efforts toward achieving educational equity.

### 1.3. Equity-Focused Positive Behavior Supports

Universal positive behavior supports are strategies that teachers use with all students (often referred to as Tier 1) to promote academic, social, emotional, and behavioral success. Prior systematic reviews of classroom management strategies and programs designed to improve social, emotional, and behavioral outcomes have identified common practices that have evidence of effectiveness in improving student performance among the samples studied [25–27]. From these reviews and in collaboration with our advisory council, we extracted 10 strategies to prioritize in the Maximize Program: (1) Personalized Greetings, (2) Student Check-ins, (3) Community Circles, (4) Establishing Classroom Expectations, (5) Acknowledging Positive Behavior (i.e., Praise), (6) Corrective Feedback, (7) Teaching Prosocial Skills, (8) Classroom Routines, (9) Effective Questioning, and (10) Student Choice. Broadly speaking, these strategies can be categorized into four interconnected domains, with some strategies applying to multiple domains: facilitating relationships (e.g., Personalized Greetings, Check-ins, Community Circles), enhancing engagement in learning (e.g., Routines, Student Choice, Effective Questions), promoting prosocial behaviors (e.g., Establishing Classroom Expectations, Acknowledging Positive Behavior, Teaching Prosocial Skills), and reducing disruptive behavior (e.g., Corrective Feedback).

There is ample evidence that these strategies facilitate a positive, welcoming, and productive classroom climate for most children [18,25,27]. However, qualitative and survey studies that highlight the educational experiences of students of color [14,28,29] and students with other marginalized identities [30,31] raise concern that current positive behavior support strategies are insufficient to achieve educational equity, as previously defined [1,2]. For example, in a study wherein adolescents of color shared their experiences of racial discrimination in school, one Black student shared the perception that, after interpersonal conflict, his teacher often accepted the explanation of the white student and required him (but not the white student) to apologize even though he did not start the altercation [29]. In another survey-based study [32], students of color and international students recounted negative experiences in which their teacher mispronounced their names, gave them "Americanized" names rather than using their own name, or connected their name to an object (to help classmates remember how to pronounce it). The informants described how these microaggressions felt disrespectful to their family and their culture, damaged the student–teacher relationship, and diminished their interest in being at school. Yet, historically, the discussion of strategies to address such experiences is rarely included in professional development trainings on traditional classroom management or positive behavioral support strategies for elementary school classrooms.

### 1.4. Guiding Framework

We used Gorski and Swalwell's (2015) Equity Literacy Framework (www.equityliteracy. org, accessed on 1 August 2022) as a foundation to inform our work in the development of

the Maximize Platform. The Equity Literacy Framework is designed to help educators not only identify quantifiable inequities in their own buildings but also understand the lived experiences of students from all backgrounds. The Equity Literacy Framework encourages educators to go beyond cultural competence and diversity awareness by developing the skill and will to engage in several critical actions, including (1) recognizing when inequities are occurring (even those that are subtle or unintentional), (2) responding to biases and inequities in the moment, and (3) redressing inequities by tackling issues that are rooted in intersecting systems of oppression [33]. According to this framework, "when we embrace equity literacy, we learn to become a threat to the existence of inequity and an active cultivator of equity in our spheres of influence" (www.equityliteracy.org, accessed on 1 August 2022). Gorski and colleagues [10,34] have written extensively on the data and experiences that have led them to develop the Equity Literacy Framework. We are unaware of any rigorous studies evaluating the extent to which the framework and its related workshops change teacher thinking and/or practice. Nonetheless, the actions and principles provide a useful framework for conceptualizing professional learning for teachers, and there is emerging qualitative evidence that experiential learning activities may facilitate growth in equity literacy skills [35].

Guided by this framework and the previously reviewed studies, we engaged in a co-creation process [36] with our advisory board members to develop equity-focused key features for each of the 10 strategies listed above [37]. The key features were designed to help teachers reduce potentially unintentional but nonetheless harmful acts and give them guidance for how to use each strategy to better serve all students in the classroom (a full list of key features for each strategy is available upon request from the first author). Some of the key features are action-oriented. For example, for Personalized Greetings, key features include (a) stating the student's preferred name, (b) offering multiple options (e.g., verbal and nonverbal; handshake, wave, bow), including an "opt out" choice (for students who may want to pass on physically interacting with a teacher), and (c) presenting the greeting options visually for students to see and select each day. For Corrective Feedback, key features include (a) considering a wide range of effective responses to disruptive behavior, including offering choices (e.g., use a take-a-break space or re-attempt the task), providing opportunities for skill development (i.e., practice the expectation that was violated), and engaging in problem-solving discussions or restorative justice conversations with peers; (b) reducing the use of consequences that exclude students from the classroom environment; (c) attempting to obtain information from all students before providing a consequence or disciplinary response; and (d) working to delay a response to challenging student behavior if the teacher is stressed, angry, or dysregulated, as this would represent a vulnerable decision point where unintended harm may occur [38]. Other features encourage self-reflection. For Acknowledging Positive Behavior, we asked teachers to reflect on the behaviors they are (un)intentionally reinforcing and prioritize behaviors that facilitate inclusivity and community building rather than only acknowledging behaviors aligned with compliance with adult authority (i.e., a Eurocentric view of "good" behavior). If teachers selected to learn more about this feature within the strategy of Acknowledging Positive Behavior, they were directed to another activity to help facilitate self-reflection on this equity topic.

### 1.5. The Maximize Platform

Given the factors known to facilitate teachers' implementation of traditional classroom management practices (e.g., performance feedback, use of data, implementation supports) [17,39], the emerging benefits of interactive technology for classroom supports [40], and the vulnerability required to engage in equity-related learning and growth, we felt there was merit in developing and evaluating a user-friendly, self-paced, interactive technology for the Maximize Program that leverages effective user engagement strategies (e.g., guided user flows, motivational elements, individualization). Given the premise that computers may facilitate honesty [41,42], we hypothesized that a private space may be particularly

useful for capturing initial reactions to equity-focused content and may be appreciated given the vulnerability needed to engage in self-reflection on topics of bias and inequity.

Several studies now demonstrate how technology-based modalities can enhance teacher implementation of standard classroom interventions [43–45]. Two past studies have demonstrated that when given access to interactive technology, a sizable percentage of teachers (39% to 51%) were able to develop a classroom intervention (i.e., a daily report card), implement it for one to two months, and produce meaningful change in student behaviors with minimal support from others [46,47]. Another evaluation of this same platform revealed that it facilitated teachers' use of several evidence-based principles for high quality classroom interventions (i.e., screening, baseline tracking, setting achievable goals, and tracking behaviors over time) [40]. Collectively, these studies demonstrate the promise of technology-based platforms for supporting teachers' implementation of positive behavior supports, and thus we felt they should be explored for the implementation of equity-focused supports. Based on these prior two studies, we hypothesized that at least 50% of teachers would engage with the Maximize Platform; however, given the sensitive nature of the topics, we also wondered if rates would be slightly lower.

Similarly, previous studies with rigorous methods assessing teachers' self-reported use of traditional positive behavior support strategies offer insights into what we might expect from teachers' self-reported use of the 10 Maximize strategies. Namely, previous studies [48–50] suggest that teachers' use of Classroom Expectations and Praise is reportedly high and that these strategies are used frequently by 70% of general and special education teachers. In contrast, fewer general education teachers (50%) report frequent use of Consequences for Behavior and Student Choice, and special education teachers report using some strategies (e.g., Student Choice in assignments) more than general education teachers do [49]. Given the overlap between the Maximize strategies and those assessed in these prior studies (e.g., Classroom Expectations, Praise, and Student Choice), we hypothesized that we would replicate these findings. However, we are unaware of studies assessing teachers' self-reported use of strategies to facilitate relationships (i.e., Personal Greetings, Check-Ins, or Community Circles) or studies that assess teachers' consideration of equity-focused features of positive behavior support strategies.

Following teacher self-assessment of strategy use, we encourage them to learn more (via Learn More pages) by exploring the equity-focused features of each strategy and to set a goal to improve one of these features. During the goal-setting process, we created elements in the user flow to enhance teacher motivation and narrow the gap between intention to implement and actual implementation of the new feature [51]. First, the Goal Builder produces a goal statement, but we allow teachers to edit it for individualization. Second, the Goal Builder requires teachers to answer two Motivational Ruler Questions [52] about the importance of their goal and their confidence in achieving this goal, as ratings on these rulers have been found to correlate with subsequent changes in teacher practices [53]. Third, the interactive Maximize Platform sends teachers a prompt to review their goal at the end of each week, as progress monitoring can contribute to changes in teacher practices [54].

### 1.6. Current Study

There is mounting interest in improving teachers' use of equity-focused positive behavior supports, but there is very little research on how to engage in this work in ways that are effective and feasible under typical practice conditions. The goal of the Maximize Program is to collaboratively partner with educators to develop procedures and materials to help teachers maximize their use of equity-focused behavior supports. In the current paper, we aim to replicate previous studies by describing elementary school educators' self-reported use of 10 positive behavior support strategies (Aim 1). We advance the literature by exploring the goals teachers set for improving equity-focused key features within each of these strategies (Aim 2) and their reported progress with regard to improvement in equity-focused practices (Aim 3) during the first quarter of the academic school year. Results

and lessons learned provide insights for research on equity-focused teacher professional development and on the use of interactive technology in this endeavor.

*1.7. Investigative Team Positionality and Epistemology*

The investigative team is led by white, cisgender women. We approach this work as allies, with humility and reflexivity. We understand that we are outsiders to many of the experiences we are working to address and that our own training and lived experiences within the dominant social group lead to biases in how we see and understand educational inequity. We engage with our advisory council as one important tool for expanding perspectives on these topics. We also acknowledge that there is a long history of practitioners and scholars of color doing this work and that we wish to build on—and not appropriate or receive undue credit for—this prior work.

Our team takes a critical stance on this work. We see educational inequities as arising from oppressive social systems [4–6] and not individual, family, school, or community deficits. We also understand all knowledge to be socially situated and created, meaning that it is subjective and relational and that there are multiple, equally valuable, and valid ways of knowing about a given topic. In our own work, this means incorporating the best-available research evidence as well as practice-based evidence and lived experiences. Our stance also means that our proposed solutions to inequities follow an anti-oppressive, social justice framework. By this, we mean that ending educational inequities requires the re-distribution of resources to ensure that resource allocation (and the process through which it is allocated) is fair and based on rights and needs [55].

## 2. Materials and Methods

*2.1. Setting and Participants*

Participants were 90 educators from three elementary schools in Central Ohio who were participating in a larger study aimed at developing tools to help educators maximize their use of equity-focused positive behavioral supports [56]. Although all staff members in each building were invited to participate, the sample for the current study was restricted to general education teachers (*n* = 55; representing 95% consents obtained for general education teachers) and other teachers (*n* = 35; representing 93% consents obtained for special education teachers, allied arts [music, art, PE] teachers, and English language learner teachers). Most educators were women (88.9%), and most identified as non-Hispanic and White (76.7%). These data suggest that our sample aligns with the national profile of teachers, in which 89% identify as women and 79% identify as White [57]. See Table 1 for participants' demographic information. According to records from the principals of the three elementary schools, the student bodies are characterized as: 18 to 25% white, 42 to 59% Black, 9 to 13% Latine, 2 to 14% Asian, and 10 to 14% mixed race. About 13 to 24% of students identify as English language learners, 14 to 17% have a disability, and 50 to 58% are eligible for free or reduced-price lunch. All three schools were actively using a PBIS framework. Each had an established set of school-wide expectations (Tier 1) for each space within the school. The staff in each building devoted significant time to teaching these expectations at the beginning of the year and as a part of a "re-set" in January. Although all three schools had designated teams for Tiers 2 and 3, they were still developing their procedures and interventions for these tiers. With regard to diversity, equity, and inclusion (DEI) initiatives, the principals reported that their staff had experienced some professional development and some staff had participated in book clubs on DEI topics, but receptivity to these efforts was mixed, and meaningful change has been slow. Demonstrable signs of efforts toward inclusivity were evidenced in the books and artwork in the hallways and teacher areas and in activities held at the schools (e.g., multi-cultural family night).

**Table 1.** Participant demographics.

| Variable * | General Education Teachers (*n* = 55) | Other Teachers (*n* = 35) | Total Sample (*n* = 90) |
|---|---|---|---|
| Position | | | |
| General education teacher | 55 (100%) | N/A | 55 (61.1%) |
| Special education teacher | NA | 26 (74.3%) | 26 (28.9%) |
| Special arts teacher (music, art, PE) | NA | 4 (11.4%) | 4 (4.4%) |
| English second language teacher | NA | 5 (14.3%) | 5 (5.6%) |
| Grade | | | |
| Kindergarten | 8 (14.5%) | N/A | 8 (8.9%) |
| 1st grade | 11 (20%) | N/A | 11 (12.2%) |
| 2nd grade | 10 (18.2%) | N/A | 10 (11.1%) |
| 3rd grade | 11 (20%) | N/A | 11 (12.2%) |
| 4th grade | 10 (18.2%) | N/A | 10 (11.1%) |
| 5th grade | 5 (9.1%) | N/A | 5 (5.6%) |
| Gender | | | |
| Man | 6 (10.9%) | 4 (11.4%) | 10 (11.1%) |
| Woman | 49 (89.1%) | 31 (88.6%) | 80 (88.9%) |
| Ethnicity | | | |
| Hispanic/Latine | 1 (1.8%) | 0 (0%) | 1 (1.1%) |
| Not Hispanic/Latino/a/x/e | 43 (78.2%) | 28 (80%) | 71 (78.9%) |
| Prefer not to answer | 0 (0%) | 0 (0%) | 1 (0%) |
| Race | | | |
| Asian/Asian American | 0 (0%) | 1 (2.9%) | 1 (1.1%) |
| Black/African American | 1 (1.8%) | 0 (0%) | 1 (1.1%) |
| White | 42 (76.4%) | 27 (77.1%) | 69 (76.7%) |
| Other [a] | 1 (1.8%) | 0 (0%) | 1 (1.1%) |
| Highest Degree | | | |
| BA/BS | 24 (43.6%) | 12 (34.3%) | 36 (40%) |
| MA/MS/Ed.M | 20 (36.4%) | 16 (45.7%) | 36 (40%) |
| Years in current position | 7.15 (7.58) | 6.39 (6.08) | 6.85 (7.00) |
| Years in current building | 7.81 (6.54) | 8.07 (6.66) | 7.91 (6.54) |
| Years in education profession | 14.0 (9.35) | 13.84 (9.49) | 13.94 (9.34) |

Note. * Percentages that do not add up to 100% represent missing data. a This participant selected "Other" as their race and, when prompted to provide their race, they did not respond. N/A indicates that the variable was not applicable for that particular group.

### 2.2. Procedures

Approval for the study was granted by the university's institutional review board and the participating school districts. To develop the first iteration of the Maximize Platform, we collaborated with a 10-member, multi-region interprofessional (i.e., general education teachers, special education teachers, administrators, diversity and inclusion leaders, school social workers, and child development specialists) advisory council. Council members were recruited via the networks of the faculty investigators. The council was comprised of women identifying as white, Black, African, Latina, children of second-generation parents, and parents of children with a disability, among other identities.

Investigators recruited schools in the winter of 2021 by distributing flyers for the project via email to elementary schools in districts located in Central Ohio. Interested principals responded to the flyer, and meetings were arranged to describe the project. Principals from three elementary schools representing two districts consented to the project. In August 2022, investigators held a project orientation meeting with staff in each building. The project was described, teachers were given the opportunity to consent, and then teachers were given the opportunity to complete a self-assessment on the Maximize Platform during the orientation (see description below). Via the backend of the platform, investigators were able to track teachers' platform use.

*2.3. Measures*

2.3.1. Technology-Based Self-Assessment

When teachers logged on to the Maximize Platform, they were directed to watch a four-minute video (available here: https://oucirs.org/2021/08/05/maximize-project/, accessed on 1 August 2022), which provided an overview of the larger project, defined equity, and described the goals and features of the platform. Immediately after the video, users were provided with traditional definitions of each of the 10 positive behavior support strategies and prompted to complete a survey inquiring about their use of each: (1) Personalized Greetings, (2) Student Check-ins, (3) Community Circles, (4) Establishing Classroom Expectations, (5) Acknowledging Positive Behavior (Praise), (6) Corrective Feedback, (7) Teaching Prosocial Skills, (8) Classroom Routines, (9) Effective Questioning, and (10) Student Choice (definitions available from the first author upon request).

For each strategy, educators read the definition of the strategy (e.g., Personalized Greetings: Each student is greeted in an individualized way when arriving in the classroom) and were asked to make two ratings: (a) frequency of use and (b) interest in improving use of this strategy. Responses for frequency of use were: *rarely* (1), *sometimes (2), half the time* (3), *often* (4), and *very often* (5). Responses for interest in improving were: *I do not use this strategy* (1), *I think I could improve my use of this strategy* (2), *I think I am doing pretty well with this strategy* (3), and *This strategy is an area of strength for me* (4). Teachers were also able to provide any additional comments on their use of each strategy via open-ended text boxes.

Based on their ratings, teachers were presented with a personalized strategy profile, which displayed each of the strategies in one of three columns: *Areas of Strength, Doing Well Enough,* and *Potential Areas for Growth* (see Figure 1). Strategies that the teacher reported using *rarely or sometimes* and/or those they reported they *could improve their use of* were assigned to the *Potential Areas for Growth* column. If the teacher reported they were *doing pretty well* combined with *rarely* or *sometimes* using the strategy, the strategy was assigned to the *Potential Area for Growth* column. If teachers reported they were *doing pretty well* with a strategy and that they used the strategy at least *half of the time*, the strategy was assigned to the *Doing Well Enough* column. If teachers reported that the strategy was an *area of strength* and reported using the strategy at least half of the time, the strategy was assigned to the *Areas of Strength* column.

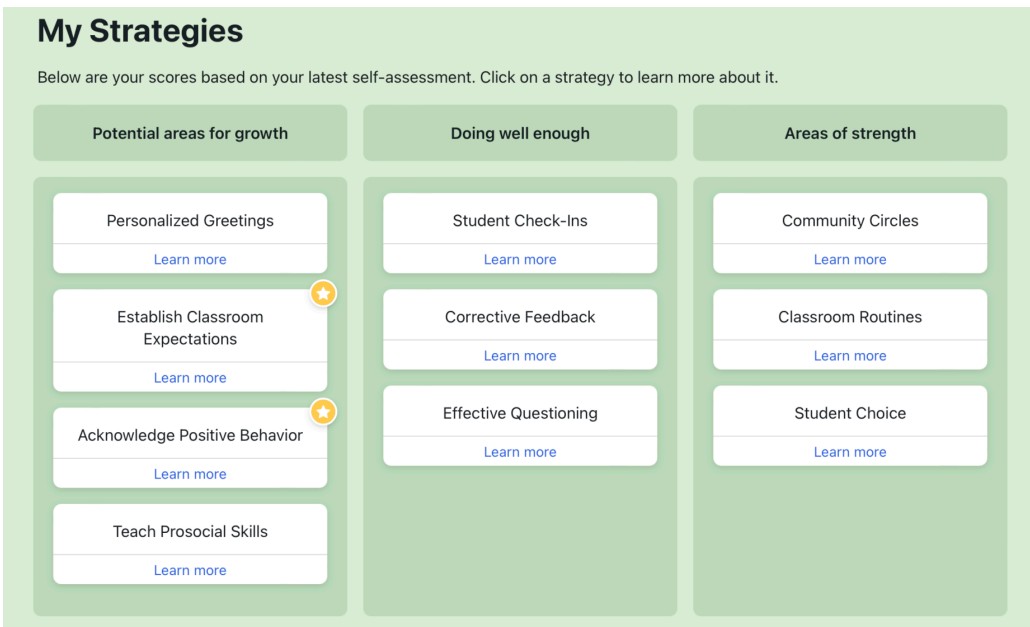

**Figure 1.** Sample teacher strategy profile produced following the self-assessments. Note. Stars indicate the strategies for which teachers set a goal for improvement.

For each strategy, teachers could click a Learn More button to be directed to a resource page for that strategy, which included the standard definition as well as key features designed to help teachers apply the strategy in an equity-focused manner. Additionally, video clips, handouts, and additional websites were provided on the resource page for each strategy so teachers could see models of the strategy and obtain materials for using the strategy (e.g., visuals for multiple greeting options).

### 2.3.2. Technology-Based Goal Setting

After completing the initial self-assessment and exploring content on the Learn More pages, teachers were encouraged to set a goal using an interactive feature (i.e., the Goal Builder). The Goal Builder first prompted teachers to select which of the 10 core strategies they wanted to improve. Then, they were prompted to select the equity-focused feature of that strategy that they wanted to improve. The Goal Builder then created a specific goal statement for this feature. For example, if a teacher chose the Corrective Feedback strategy and selected the equity-focused feature for improvement, "Reducing the use of consequences that exclude children from the classroom", the Goal Builder auto-populated the following statement: I am improving my use of corrective feedback by reducing the use of consequences that exclude children from the classroom. Teachers were allowed to accept this goal statement or edit it for individualization.

Lastly, consistent with principles of motivating change behavior [52], educators were asked to complete Motivational Ruler ratings for importance and confidence related to their goal (i.e., Among all other things you have to do, how important is this goal? How confident are you that you can improve this practice in the next week?). Responses to these two statements were on a 10-point scale, from 1 = *not at all important/confident* to 10 = *very important/confident*. Once teachers completed the Goal Builder, they were redirected to their dashboard, where they could see their goal statement each time they logged onto the platform.

### 2.3.3. Technology-Based Goal Review

At the end of each week, teachers received a prompt on the platform to review their goals. First, they were asked: How have you done with your goal since your last review date? Response options were: (1) *Oops I forgot*; (2) *I made a little progress*; or (3) *I made a lot of progress*. If they selected *Oops I forgot*, the Goal Builder inquired about the barriers that got in the way and what the teacher might do to work on them in the next week. If they selected *I made a lot of progress*, the Goal Builder asked them to describe the progress they had made. Regardless of their first response, they were then asked: What do you want to do with this goal? Response options were: (1) *Stop working on it*; (2) *Modify it and keep working on it*; (3) *Keep working on it with no changes*; and (4) *I've mastered this one! Let's consider it an achievement*. If they selected *I've mastered this one* prior to a month's time elapsing, they were encouraged to keep working on it for 4 weeks in order to make the practice a habit. If they selected this option after a month had passed and they had completed two goal reviews, then they were asked with whom they might share this success (e.g., with their principal, mentor, or colleagues) to promote sharing and celebration.

### 2.3.4. Data Analysis

For Aim 1, we report descriptive statistics highlighting teachers' self-reported use of each strategy and their perceptions of areas of strength, doing well enough, and areas for potential growth. Because our sample included both general education teachers and educators in other roles, we used Chi-Square analyses to explore the patterns of response by educator type to be able to compare our results to other studies [49]. For Aim 2, we report which equity-focused features teachers selected as goals for improvement and the percent of teachers who selected each one. Given the autonomy provided to teachers on the platform, we were interested in the features that were selected for improvement, as these features suggest aspects of practices that teachers are willing to change and may highlight practices to prioritize for future implementation support. For Aim 3, we report the percent

of teachers who reviewed their goals and their perceptions of progress toward these goals. These data indicate the extent to which the interactive platform facilitates teacher progress monitoring and reveals how teachers assess their own progress toward improvement.

## 3. Results

*3.1. Aim 1: Teacher Self-Reported Use and Interest in Improving Strategies*

Table 2 shows the percent of teachers who reported using each strategy *never/rarely*, *half of the time,* and *often/very often*. More than 70% of general education teachers *and* more than 70% of other teachers (i.e., special education, allied arts) reported using these four strategies *often/very often*: Personalized Greetings, Classroom Expectations, Acknowledging Positive Behavior, and Routines. For other strategies, the patterns differed by teacher type. Namely, chi-square tests revealed significant proportional differences between general education teachers and other teachers for Community Circles ($X^2(2, N = 90) = 38.30$, $p < 0.001$); the standardized residuals indicated that a greater proportion of general education teachers endorsed frequent use of Community Circles. Chi-square tests also revealed significant proportional differences for Corrective Feedback ($X^2(2, N = 90) = 6.88$, $p = 0.032$) and Effective Questioning ($X^2(2, N = 90) = 6.74$, $p = 0.034$). Although the standardized residuals did not exceed $+/-2$ for any cell, the pattern in Table 2 suggests that a greater proportion of other teachers endorsed frequent use of these strategies.

**Table 2.** Frequency of strategy use as reported by teachers (collapsed).

| Maximize Strategy | Never/Rarely | Half of the Time | Often/Very Often |
|---|---|---|---|
| **Total Sample (N = 90)** | | | |
| Greetings | 11.1% | 12.2% | 76.7% |
| Check-Ins | 11.1% | 25.6% | 63.4% |
| Community Circles * | 33.3% | 11.1% | 55.6% |
| Classroom Expectations | 5.5% | 7.8% | 86.6% |
| Acknowledge Pos. Behavior | 2.2% | 12.2% | 85.5% |
| Corrective Feedback * | 4.4% | 32.3% | 63.3% |
| Teaching Prosocial | 10.0% | 22.2% | 67.8% |
| Routines | 2.2% | 6.7% | 91.1% |
| Effective Questioning * | 11.1% | 32.2% | 56.6% |
| Student Choice | 20.0% | 33.3% | 46.7% |
| **General Education Classroom Teachers (n = 55)** | | | |
| Greetings | 7.2% | 12.7% | 80.0% |
| Check-Ins | 10.9% | 30.9% | 58.1% |
| Community Circles | 9.1% | 12.7% | 78.1% |
| Classroom Expectations | 3.6% | 5.5% | 90.9% |
| Acknowledge Pos. Behavior | 3.6% | 14.5% | 81.8% |
| Corrective Feedback | 5.5% | 41.8% | 52.7% |
| Teaching Prosocial | 9.1% | 21.8% | 69.1% |
| Routines | 1.8% | 5.5% | 92.7% |
| Effective Questioning | 7.2% | 41.8% | 50.9% |
| Student Choice | 21.8% | 30.9% | 47.3% |
| **Other Teachers (n = 35)** | | | |
| Greetings | 17.1% | 11.4% | 71.4% |
| Check-Ins | 11.4% | 17.1% | 71.5% |
| Community Circles | 71.5% | 8.6% | 20.0% |
| Classroom Expectations | 8.6% | 11.4% | 80.0% |
| Acknowledge Pos. Behavior | 0.0% | 8.6% | 91.4% |
| Corrective Feedback | 2.9% | 17.1% | 80.0% |
| Teaching Prosocial | 11.5% | 22.9% | 65.7% |
| Routines | 2.9% | 8.6% | 88.6% |
| Effective Questioning | 17.2% | 17.1% | 65.8% |
| Student Choice | 17.1% | 37.1% | 41.7% |

Note. Other teachers include special education teachers, allied arts teachers (physical education, music, and art), and English language learning teachers. * Indicates a significant difference in the pattern of responses by general education and other teachers.

Table 3 shows the percent of teachers who rated each strategy as an area of strength or area of growth with interest in improving the strategy. Among general education teachers, the strategies most often rated as strengths (>30% of teachers) were Classroom Expectations and Routines. Approximately 30% to 50% of general education teachers reported that they "could improve" their use of most strategies, with the exception of Personalized Greetings and Routines. Among other teachers, the strategy most often rated (>30%) as a strength was Acknowledging Positive Behavior. Approximately 30% to 50% of other teachers reported that they could improve their use of all other strategies.

**Table 3.** Self-perceptions of strengths and areas for growth as reported by teachers.

| Maximize Strategy | Do Not Use | Can Improve | Doing Pretty Well | Area of Strength |
|---|---|---|---|---|
| **Total Sample (*N* = 90)** | | | | |
| Greetings | 3.3% | 23.3% | 48.9% | 24.4% |
| Check-Ins | 2.2% | 46.7% | 38.9% | 12.2% |
| Community Circles | 13.3% | 42.2% | 27.8% | 16.7% |
| Classroom Expectations | 1.1% | 32.2% | 36.7% | 30% |
| Acknowledge Pos. Behavior | 0% | 26.7% | 40.0% | 33.3% |
| Corrective Feedback | 0% | 45.6% | 42.2% | 12.2% |
| Teaching Prosocial | 1.1% | 46.7% | 41.1% | 11.1% |
| Routines | 1.1% | 30.0% | 32.2% | 36.7% |
| Effective Questioning | 3.3% | 50.0% | 37.8% | 8.9% |
| Student Choice | 2.2% | 56.7% | 32.2% | 8.9% |
| **General Education Classroom Teachers (*n* = 55)** | | | | |
| Greetings | 3.6% | 20.0% | 52.7% | 23.6% |
| Check-Ins | 0% | 52.7% | 38.2% | 9.1% |
| Community Circles | 0% | 36.4% | 43.6% | 20% |
| Classroom Expectations | 0% | 30.9% | 34.5% | 34.5% |
| Acknowledge Pos. Behavior | 0% | 30.9% | 45.5% | 23.6% |
| Corrective Feedback | 0% | 49.1% | 38.2% | 12.7% |
| Teaching Prosocial | 0% | 49.1% | 40% | 10.9% |
| Routines | 0% | 29.1% | 27.3% | 43.6% |
| Effective Questioning | 3.6% | 52.7% | 36.4% | 7.3% |
| Student Choice | 0% | 56.4% | 36.4% | 7.3% |
| **Other Teachers (*n* = 35)** | | | | |
| Greetings | 2.9% | 28.6% | 42.9% | 25.7% |
| Check-Ins | 5.7% | 27.1% | 40% | 17.1% |
| Community Circles | 34.3% | 51.4% | 2.9% | 11.4% |
| Classroom Expectations | 2.9% | 34.3% | 40% | 22.9% |
| Acknowledge Pos. Behavior | 0% | 20% | 31.4% | 48.6% |
| Corrective Feedback | 0% | 40% | 48.6% | 11.4% |
| Teaching Prosocial | 2.9% | 42.9% | 42.9% | 11.4% |
| Routines | 2.9% | 31.4% | 40% | 25.7% |
| Effective Questioning | 2.9% | 45.7% | 40% | 11.4% |
| Student Choice | 5.7% | 57.1% | 25.1% | 11.4% |

### 3.2. Aim 2: Teacher Goal Setting for Equity-Focused Implementation

Of the total sample, 59 out of 90 teachers set a goal following the completion of their self-assessment (67% of general education teachers and 62% of other teachers). Teachers who set a goal (*n* = 59) did not differ from teachers who did not set a goal (*n* = 31) with regard to gender or race, and goal setting by teachers across schools was proportionate to the sample sizes across schools. When selecting their goal, 76.3% of participants selected a strategy that was listed as a *Potential Area for Growth* on their profile; 22% chose a strategy that was in the *Doing Well Enough* category; and one person selected a strategy that was in their *Area of Strength* category. Among general education teachers who set a goal, 18.9% set a goal for Student Choice, 16.2% for Corrective Feedback, 13.5% for Student Check-ins, 11% for Community Circles, and 11% for Acknowledging Positive Behavior, with the remaining

goals dispersed across the other strategies. Among other teachers, 18.2% set a goal for Corrective Feedback, 14% for Personalized Greetings, 14% for Community Circles, and 14% for Routines, with the remaining goals dispersed across the other strategies. Table 4 highlights the variability in the equity-focused features selected for improvement within each strategy. The most frequently selected equity-focused features were: using community circles to learn about each other's families, strengths, and talents ($n = 4$); using a wide range of effective responses to disruptive behavior ($n = 3$); helping students see how social-emotional learning (SEL) skills can be used to create social change ($n = 4$); adjusting the type of opportunities to respond (OTRs) to match student needs ($n = 3$); and offering choice across a variety of activities each week ($n = 7$).

**Table 4.** Goal setting by educators ($n = 59$).

| Strategy | Equity-Centered Features Focused on in the Goal Setting Process |
|---|---|
| Greetings ($n = 5$) | Visually presenting multiple options with an opt out choice ($n = 2$);<br>Including statement that communicates the student is welcomed/valued ($n = 2$);<br>Greet every student at least once per day ($n = 1$) |
| Check-Ins ($n = 7$) | Using Check-Ins to assess students' emotions ($n = 2$);<br>Using a tracking system to ensure all students receive a Check-In ($n = 1$);<br>Pairing Check-Ins with Greetings ($n = 2$);<br>Integrating Check-Ins throughout the day ($n = 2$) |
| Community Circles ($n = 7$) | Using Community Circles to learn about each other (family, strength, talent, etc.) ($n = 4$);<br>Holding them at a consistent time ($n = 1$);<br>Using Community Circles for prosocial skill development and problem solving ($n = 1$);<br>Asking students to develop topics and agendas for Community Circles ($n = 1$) |
| Classroom Expectations ($n = 3$) | Reviewing posted expectations prior to the start of most activities ($n = 1$);<br>Helping students practice behavioral expectations (photos/draw/write them) ($n = 2$) |
| Acknowledge Pos. Behavior ($n = 5$) | Using practice statements strategically (change behavior, improve relationships, etc.) ($n = 2$);<br>Reflecting on biases in my use of praise ($n = 1$);<br>Ensuring all students receive praise every day ($n = 2$) |
| Corrective Feedback ($n = 10$) | Using a wide range of effective responses to disruptive behavior ($n = 3$);<br>Understanding key times for ignoring student behavior ($n = 1$);<br>Slowing down before issuing a consequence when I am stressed ($n = 2$);<br>Reducing the use of consequences that exclude students from the classroom ($n = 2$);<br>Identifying and modifying possible triggers to disruptive student behavior ($n = 1$);<br>Examining data about which students are and are not receiving consequences ($n = 1$) |
| Teaching Prosocial ($n = 6$) | Offer opportunities for all students to practice social-emotional learning (SEL) skills ($n = 1$)<br>Helping students see how SEL skills can be used to create social change ($n = 4$)<br>Communicating with caregivers about SEL skills taught and ask for input ($n = 1$) |
| Routines ($n = 6$) | Reteaching or revising routines at key times throughout the year ($n = 2$)<br>Posting visual depictions of routines that are reflective of classroom ($n = 1$)<br>Initiating routines using language that respects and celebrates all ($n = 1$)<br>Co-developing routines with students to enhance buy-in and engagement ($n = 2$) |
| Effective Questioning ($n = 3$) | Use opportunities-to-respond (OTRs) strategically to assess if students are attending ($n = 1$)<br>Ensuring all students are called upon (using randomized strategies) ($n = 1$)<br>Adjusting type of OTRs to match student needs ($n = 3$) |
| Student Choice ($n = 10$) | Reinforcing use of prosocial skills within the process of Student Choice ($n = 1$)<br>Offering choice across a variety of activities each week ($n = 7$)<br>Depicting choice options in auditory and visual formats and including a 'teacher choice' for those who are uncomfortable making a choice ($n = 2$) |

### 3.3. Aim 3: Progress on Goals

Of the 59 teachers who set a goal, 25 completed at least one goal review (8 people completed 1 goal review, 9 people completed 2 goal reviews, 6 people completed 3 goal reviews, and 2 people completed 4 goal reviews) in the first quarter of the year. Thirty-

four teachers who set a goal did not complete a goal review. Of the 8 teachers who completed only one goal review, 12.5% responded *Oops! I forgot*; 62.5% reported *making a little progress*; and 25% reported *making a lot of progress.* For teachers that only completed one goal review, 87.5% reported wanting to keep working on the goal with no changes, whereas 12.5% reported wanting to modify the goal and keep working on it. Barriers reported by those teachers who reported *Oops I forgot* included the general demands of the beginning of the year, difficulties with time management, and severe student behaviors that distracted teachers from their goals. Progress made by those who reported *making a lot of progress* included intentionally making plans for strategy use and keeping personal notes or checklists to track how often they used the strategy.

For those who completed multiple goal reviews, teachers responded *Oops! I forgot* 9.1% of the time, reported *making a little progress* on their goal 56.8% of the time, and reported *making a lot of* progress on their goal 34.1% of the time. Of those who did multiple goal reviews, two teachers reported mastering a goal. Teachers who completed multiple goal reviews reported wanting to keep working on the goal with no changes 84.1% of the time, whereas teachers reported wanting to modify and continue working on the goal 11.4% of the time.

When teachers initially created goals, they rated how important the goal was to them among all the other things they had to do, as well as how confident they were that they could improve their practice in the next week. An independent sample t-test was conducted to determine if there were differences in importance and confidence ratings between teachers that completed at least one goal review and teachers that did not complete a goal review. Contrary to expectations, there was no statistically significant difference in importance ratings between goal review completers ($M = 7.79$, $SD = 1.77$) and non-completers ($M = 8.03$, $SD = 1.34$), and there was also no significant difference in confidence ratings between goal review completers ($M = 7.38$, $SD = 1.72$) and non-completers ($M = 6.74$, $SD = 1.91$).

## 4. Discussion

In the Maximize Program, we are building on the strengths of the PBIS movement *and* encouraging teachers to reflect on equity-focused modifications to common PBIS strategies. In this study, we evaluated the extent to which we could leverage interactive technology to provide a private space for self-reflection, self-assessment, goal setting, and progress monitoring. We prioritized teacher autonomy in the process and gradually titrated the equity-focused content, given that previous receptivity to equity-focused work in participating schools was reportedly mixed. Below, we reflect on our findings in the context of existing literature and future research.

### 4.1. Teacher Self-Assessment

Consistent with previous literature [48–50], we found that more than 70% of general education and other teachers in our sample reported frequent use of Classroom Expectations and Praise when provided with traditional definitions of these strategies. These replicated findings are encouraging, as these are important strategies for enhancing student success in the classroom. Their common use may also reflect the positive impact of state-mandated PBIS trainings that have occurred across the United States over the last decade.

In addition, this study contributes to new findings by examining teachers' self-reported use of relationship-focused classroom strategies. Most teachers in our sample reported frequent use of Greetings and Community Circles and moderate use of Check-Ins. These results are also encouraging, as student-teacher relationships and peer relationships create a strong foundation for students feeling connected to school and facilitate engagement in the learning process [18,58], which are crucial mediators to desired academic and social outcomes for all students.

Consistent with previous studies, general education teachers reported more frequent use of Classroom Expectations and Praise than Student Choice, Effective Questioning, and Corrective Feedback [48–50]. Although non-general education teachers reported using

Effective Questioning more frequently than general education teachers, 17% of non-general education teachers still reported using this strategy rarely. Given their overall low usage as well as the importance of effective questions in facilitating engagement and autonomy in learning [25], additional research is warranted to understand possible barriers to use of this strategy (e.g., limited knowledge of the practice, comfort in using the practice, implementation planning, and supports), as this could guide solutions to enhancing use.

Lastly, the results suggest that non-general education teachers reported using Corrective Feedback more than general education teachers. This may be because special education teachers interact more with students who need more support and guidance with self-regulation skills, or it may be that general education teachers receive less training in this critical area. Providing feedback in a way that preserves student dignity and prevents the escalation of disruptive behavior is important to reducing challenging behavior in the classroom [20]. In addition, situations that require teachers to give corrective feedback are often stressful and are impacted by teacher bias, which in turn contributes to discipline disparities [38,59,60]. Given the known link between exclusionary disciplinary practices and subsequent negative outcomes (i.e., the school-to-prison pipeline [61]), particularly for Black students, Corrective Feedback is an important target for professional development related to equity-focused positive behavior supports, as well as additional research to understand barriers to use and solutions to address those specific barriers. Indeed, nearly 50% of general education teachers in our sample indicated that they could improve this strategy, and 10% of the goals that were set were focused on this strategy. Collectively, these results may suggest teachers' willingness and receptivity to such professional development.

*4.2. Teacher Goal Setting for Equity-Focused Practices*

As for goal setting, based on previous studies of teachers using interactive technology for classroom interventions [46,47], we anticipated that 39 to 51% of teachers would use the technology to set goals and monitor progress toward their goals. Within the first quarter of the year, 67% of general education teachers and 62% of other teachers set a goal for improvement. These rates may be higher than expected for several reasons. First, teachers completed the initial self-assessment during a structured time (Maximize Program orientation meeting), which may have enhanced teachers' perceptions of the priority of the activity (i.e., supported by their administrators). Perhaps this also facilitated their return to the platform to set a goal after the meeting. Second, teachers received contact hours for completing activities on the platform. Third, engagement in these activities was likely less burdensome than those in previous studies [46,47] where interactive technology was used to help teachers implement a targeted intervention with a student over time.

This is the first study to assess teachers' goal-setting as it relates to equity-focused features of a given positive behavioral support strategy. Most teachers also selected a strategy to work on from their *Areas for Growth* profile, suggesting that they generally accepted or agreed with the profile produced and likely used this profile as a means for self-reflection and goal setting. Results also indicate that there was much diversity in the strategies that teachers chose to focus on for improvement and goal setting. That is, there was no clear, universally preferred strategy for goal setting. Further, the data in Table 4 suggest that there is variability in the equity-focused features selected for improvement. On one hand, this may highlight the complex road ahead for professional development in equity-focused implementation (i.e., there are many features to work on to achieve the desired outcomes). On the other hand, variability may highlight an important lesson about allowing teachers to choose the feature they most want to work on as an important step toward behavioral change. The array of strategies that teachers selected also underscores the importance of autonomy and choice within the interactive platform. We encourage researchers to continue to explore the role of autonomy and teacher choice in the process of professional development regarding equitable practices, particularly how this relates to the actual implementation of equity-focused strategies in the classroom.

Most teachers reported that their goals were important to them and that they felt rather confident in their ability to improve their practice (i.e., scores of 7 or higher out of 10 on the Motivation Rulers). These high ratings of importance and confidence may again reflect that allowing teachers to self-identify goals may be more welcoming and motivating than a top-down approach where evaluators select goals for them. However, unlike in previous studies that used Motivational Rulers with teachers [53], these ratings were not associated with subsequent teacher behavior change. We had hoped that embedding interactive motivational interviewing techniques (i.e., ruler ratings) in the platform would facilitate motivation; however, such facilitation may not happen in a private space, as motivational interviewing theory suggests that stating change talk aloud to others is a key mechanism of action [52]. To this end, in the broader Maximize Program, we are also evaluating the extent to which connections with peer leaders [62] may further support teacher improvement in the goals they set.

It is also important to note, though, that there was a small subsample of teachers who reported low to moderate confidence in making progress on their goal within a week (6.8% reporting low confidence and 28.8% reporting moderate confidence). These are important data to collect and track because teachers reporting low confidence in their ability to improve their practices likely would benefit from additional consultation from colleagues or coaches to help them improve their equity-focused practices.

### 4.3. Goal Review and Progress Monitoring

About 30% of the teachers who set an equity-focused goal only completed one goal review in the first quarter of the academic year. During their goal review, most of these teachers indicated they had forgotten about the goal (12.5%) or had made little progress (62.5%). Barriers reported by these teachers aligned with extant literature on barriers to implementation (i.e., competing demands, time management, stressful student behavior) [63,64]. As described above, data such as these (i.e., the absence of progress monitoring or goal reviews) could be used to determine which teachers may need interpersonal support in addition to interactive technology to make a change in practices.

However, two-thirds of teachers who set a goal went on to use the Maximize Platform for multiple goal reviews. These numbers are encouraging, as they suggest that teachers did return to the platform for progress monitoring and self-reflection. Facilitators of progress (as reported by teachers in the goal reviews) included intentionally making plans for strategy use and tracking strategy use. This resonates with other findings that intentionality, implementation planning, and progress monitoring are important levers for behavior change [63,64]. Thus, future studies should include a measure of observed behavior change and its potential connection to these reported facilitators. Many teachers who completed multiple goal reviews reported wanting to continue working on the goal without making changes. Variability in responses for how much progress teachers had made on their goals in the past week (including the *Oops! I forgot* responses) suggests some honesty in responses, the importance of normalizing slow progress, and likely highlights that some teachers need additional interpersonal supports (e.g., a coach or consultant) to move from intention to action.

Despite the encouraging number of teachers who set a goal, the number of teachers who engaged in the platform declined with each respective step of the technology-driven process (e.g., initial, then subsequent, goal reviews). Thus, although teachers may initially engage, we need to examine mechanisms for sustained engagement in the context of real-world barriers, such as teachers' busy daily schedules. To understand one possible mechanism, we are currently examining the extent to which teachers and staff who are viewed as leaders within the building can serve as an interpersonal catalyst for engagement both in the technology platform and in deeper in-person discussions about equity-focused classroom practices.

*4.4. Limitations*

While this study has strengths, it is not without limitations. First, because we did not systematically observe teachers, we are not able to report on whether teachers' self-reported use of practices via the self-assessment aligns with observed practices in the classroom. Anecdotally, in some cases, teachers reported they may have been too "hard on themselves" during their first self-assessment and less willing to report on or acknowledge their strengths. On the other hand, there is evidence that teacher self-report is more generous or positive than observed behaviors and thus may overrepresent teaching practices [65]. Thus, it is unclear to what extent the self-reported use of practices in this sample overrepresents or underrepresents teachers' actual use of practices. Relatedly, although we collaborated with an advisory council that holds expertise in culturally sustaining practices, the equity-focused features that we provided to teachers likely do not represent all the necessary features for achieving equitable outcomes. Indeed, defining equitable practices is an emerging science [66,67]. Nonetheless, the features selected suggest which strategies and key features, of those available, teachers view as important and ones they are willing to work on. Finally, in this first iteration of the platform, we specifically used traditional definitions of the strategies during the initial self-assessment while encouraging teachers to navigate to the Learn More pages for additional resources on equity-focused features. We did this because we were not sure how "ready" this audience was for the equity-focused features, and we wanted to highlight them as something unique and separate from the traditional definitions. However, in doing so, we did not collect teacher self-assessment responses on all equity-focused features specifically. Thus, as we are currently developing the second iteration of the Maximize Platform, we are embedding equity-focused features within the initial self-assessment rather than just resources within the Learn More pages or key features to choose from in the Goal Builder. This will allow us to prioritize equity during teachers' initial interactions with the platform and will also allow us to obtain a report of teacher use of all equity-focused features for each strategy.

## 5. Conclusions

The findings of this study replicate previous research on teachers' self-reported use of traditional positive behavior supports. Our findings also extend our understanding of teachers' self-reported use of strategies in new domains (i.e., Facilitating Relationships), the individualized goals they set for improving equity-focused practices related to these traditional strategies, and the extent to which interactive technology can be used to facilitate goal setting, goal review, and motivation towards achieving equity-focused goals.

Findings from the current study also expand upon the literature on equity-focused positive behavioral supports in a few major ways. First, our results highlight strategies that are frequently and comfortably used by teachers (i.e., frequent use of Classroom Expectations and Praise) while also suggesting areas for additional focus in professional development and additional research on barriers to implementation (i.e., Corrective Feedback, Effective Questioning, and Student Choice). Given that we hypothesize that improving the use of these strategies and their key features is related to reducing disproportionality in discipline, this is a promising finding. Second, our findings offer optimism about teachers' openness to growth in using equity-focused features in that, despite the everyday demands of teaching, over half of the teachers in our sample took the time to set a goal with the intention of improving equity-focused positive behavior supports in their classroom. Third, our findings suggest that some teachers are willing to use interactive technology that offers a private space where they can engage in self-reflection and goal-setting that may facilitate equitable outcomes for their students. We will triangulate the quantitative data from this study with the qualitative data we are gaining from key informant interviews with high and low users of the platform (about what features teachers do and do not find useful) to guide future modifications of the platform. Lastly, results suggest that with each step on the platform, engagement wanes. Thus, we will be exploring the extent to which highly

regarded peers, school-based behavioral specialists, and equity leaders can further serve as interpersonal catalysts for engaging teachers in this important work.

Inequities in educational experiences and outcomes remain, and change is needed at the federal, state, and local levels to address these disparities. Within the Maximize Program, we are looking for feasible and effective ways that teachers can make an equity impact within their classroom. Our results offer insights for researchers and practitioners into teachers' perceived use of equity-centered classroom management practices as well as how we can leverage interactive self-paced technology to move the needle toward teachers' implementation of equity-focused practices.

**Author Contributions:** Conceptualization, J.S.O., D.E.-C. and E.C.; Methodology, J.S.O., M.D., J.S., E.C. and N.M.; Software, J.S.O., M.D., J.S. and N.Z.; Validation, J.S.O.; Formal analysis, J.S.O.; Data curation, J.S.O., M.D., J.S. and N.Z.; Writing—original draft, J.S.O., D.E.-C., M.D. and J.S.; Writing—review & editing, E.C., N.M. and N.Z.; Supervision, E.C.; Project administration, J.S.O. and D.E.-C.; Funding acquisition, J.S.O., D.E.-C. and E.C. All authors have read and agreed to the published version of the manuscript.

**Funding:** The research reported here was supported by the Institute of Education Sciences, U.S. Department of Education, through Grant R305A210224 to Ohio University. The opinions expressed are those of the authors and do not represent views of the Institute or the U.S. Department of Education.

**Institutional Review Board Statement:** The study was conducted in accordance with the Declaration of Helsinki and approved by the Institutional Review Board (or Ethics Committee) of Ohio University (protocol code 21-F-24; approved 6/28/21) for studies involving humans.

**Informed Consent Statement:** Informed consent was obtained from all subjects involved in the study.

**Data Availability Statement:** Data are available upon request from the first author. The full dataset will become publicly available after the end of the trial.

**Acknowledgments:** We extend our gratitude to all members of the Maximize Project Advisory Council for their generosity in sharing their time, expertise, and perspectives with us. We appreciate the meaningful contributions they made to the development and refinements of the products and resources associated with this project. This council is comprised of experts in their field who represent a diversity of lived experiences, student populations, and geographic locations across North America. We also thank all participating teachers for their contributions.

**Conflicts of Interest:** The authors declare no conflict of interest.

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
