# Peer review of "Elementary School Teachers’ Self-Assessment of Use of Positive Behavior Support Strategies and Goal Setting Related to Equity-Focused Features"

_education, doi:10.3390/educsci13080847_

Round 1

Reviewer 1 Report

Thank you for the opportunity to review this manuscript, which examined classroom teachers’ use of an online platform to learn about, set goals, and monitor their progress in equity-focused behavior supports. I think this topic is incredibly important for the field, and any attempts to learn how to increase equity in teacher-student interactions is to be commended, especially as it pertains to intervention research. However, I believe there are some edits that would be worth considering. My concerns and suggestions for improvement are as follows:

1.      Abstract. The abstract notes three strategies but lists four.

2.      Equity literacy framework. Is there any research on this approach to equity?

3.      Strategy materials. I think it would be helpful to the reader to have more details on how the strategies (and the equity enhancements) were developed. The authors mention some involvement of an advisory board, but more information or reference to a published paper would help.

4.      Measure of teacher interest. Not all of the response options are provided.

5.      Personalized profile. What feedback is provided if teachers rate themselves as doing well and having no areas of interest?

6.      Goal setting. Can users set a goal that’s not an equity-enhancement goal?

7.      Figure 1. The example doesn’t include all 10 strategies. How are these (not) presented to users?

8.      Participating school characteristics. What information can the authors provide about the schools themselves (e.g., school characteristics)? What other initiatives (e.g., PBIS, equity work) were happening in the schools or districts?

9.      Analysis section. I recommend that the authors include a Data Analysis section at the end of the Method to help the reader follow the results. This section should state the procedures and provide rationales for the authors’ analytic decisions.

10.  Table 2. I’d prefer to see the raw (i.e., uncollapsed) results.

11.  Chi-square results. Technically, chi-square results indicate proportional differences but not where the greater proportions are.

12.  Discussion. How do we know the extent to which teachers actually set an EQUITY goal?

13.  Additional data. Related to the previous point, is it possible to obtain counts of whether (and which) teachers clicked the learn more button to explore equity? I think this would be a really useful additional bit of data to include!

14.  Progress results. Include the n of those who never completed a goal review on p. 14).

15.  Nesting. Were there any school-level differences in results that should be accounted for in the analyses?

In sum, I think the authors have done excellent work in developing this platform and applying it to equity in school discipline. I wish them all the best in continued study regarding their line of research.

Author Response

Response to Editor Comments

Point 1: It is clear the authors have a social justice bias, which is fine, but it is really obvious. Line 50, I would suggest you try to tone down the rhetoric; especially when you provide no substantiation for these opinions. In fact, there were several instances where the authors provide an opinion or statement of presumed fact without substantiating their claims (e.g., line 230).

Response 1: Thank you for this feedback. We added a positionality and epistemology statement to make our social justice lens explicit. We have also reviewed the article carefully and added citations to substantiate claims where needed.

Point 2: I agree with reviewer 1 that self-report survey data can be problematic. For example, I would have liked it if the authors had followed up with teachers to find out why some interventions were supported by teachers and others not. While the chi squared analysis is appropriate for determining that the proportions were different, when it comes to teachers reporting they often used specific strategies, of more importance in term of research would be to find out why teachers didn’t choose to implement specific strategies. As the authors note on line 522 and 556, a weakness of this study is the lack of follow up with teacher. You know what they said they did but you haven’t verified changes in behavior, relying on teacher self-report. This really limits the value of the study.

Response 2: Thank you for this feedback. In the first iteration of using the platform, we focused on teachers self-report. We agree that the current data limit our ability to draw conclusions about teacher behavior change. We reviewed the article and added in clarifying statements to ensure that the reader is clear this is about teacher perceptions. We also modified our results about the teacher feedback in order to ensure that we are not overstating our conclusions regarding these data. Finally, we added text in the discussion that mentions following up with teachers as a future direction for this work.

Point 3: In the conclusions, the authors state on line 600 that the results of this study provide insights into teachers’ willingness to use interactive technology, but they don’t say what those insights were. They say they were optimistic about teachers’ willingness to support equity-based strategies but the data doesn’t really support this. The authors also state that they have highlighted strategies teachers were interested in using suggesting that more PD is need to get teachers interested in other less enticing strategies.  Training is not always the solution. The authors would first need to find out why the strategies are unappealing to teachers. Also, why there interest diminishes. And lastly, the recommendation on line 602 to encourage more research is on teacher autonomy should be place in a future research section.

Response 3: We modified text about insights. We clarified areas of optimism and caution. We modified our comments related to PD. Rather than separating future research into a new section, which can read as disjointed, we describe future research ideas within each section of the discussion.

Response to Reviewer 1 Comments

Point 1: Abstract. The abstract notes three strategies but lists four.

Response 1: Thank you for catching this typo. We changed “three” to “four”

Point 2: Equity literacy framework. Is there any research on this approach to equity?

Response 2: Gorski and colleagues have written extensively on the data and experiences that have led them to develop the Equity Literacy framework, which serves as a grounding theoretical model for our work. Rigorous empirical research evaluating the extent to which the framework and its related workshops change teacher thinking and/or practice is limited. We now acknowledge this and have added references highlighting emerging research that has attempted to evaluate change in equity literacy practices.

Bukko, D., & Liu, K. (2021, March). Developing preservice teachers’ equity consciousness and equity literacy. In Frontiers in Education (Vol. 6, p. 586708). Frontiers Media SA.

Point 3:  Strategy materials. I think it would be helpful to the reader to have more details on how the strategies (and the equity enhancements) were developed. The authors mention some involvement of an advisory board, but more information or reference to a published paper would help.

Response 3: Thank you for your interest in this. Given the in-depth and iterative nature of this work, these details are shared in a separate paper focused on our advisory board co-creation process (to be submitted at the end of August). However, we added some details to the current paper to address this request.

Point 4: Measure of teacher interest. Not all of the response options are provided.

Response 4: With respect, we believe all response options are provided. Please let us know if we have misunderstood the request.

Point 5: Personalized profile. What feedback is provided if teachers rate themselves as doing well and having no areas of interest?

Response 5: The algorithm that produces the personalized profile is based on the combination of the frequency of use question (rarely, sometimes, half the time, often, and very often) and teachers’ report of interest (do not use, I think I could improve, doing well enough, area of strength). If the teacher reported they were doing pretty well combined with rarely or sometimes using the strategy, the strategy was assigned to the Potential Area for Growth column. We added this combination in the manuscript. Algorithmically it is possible for a teacher profile to show no strategies under Potential Areas for Growth (i.e., all strategies fell in the other two columns); there were 3 teachers in the sample who has such a profile. However, they could still review the equity-focused features of any strategy (e.g., one in the Doing Well Enough category) and create a goal for that strategy and feature.

Point 6: Goal setting. Can users set a goal that’s not an equity-enhancement goal?

Response 6: No. As described on line 330, when setting a goal, teachers were prompted to select the equity-focused feature of that strategy that they wanted to improve.

Point 7: Figure 1. The example doesn’t include all 10 strategies. How are these (not) presented to users?

Response 7: Thank you for pointing this out. This was just a portion of the screenshot. We have added a new figure so that all 10 strategies are visible. 

Point 8: Participating school characteristics. What information can the authors provide about the schools themselves (e.g., school characteristics)? What other initiatives (e.g., PBIS, equity work) were happening in the schools or districts?

Response 8: We added the requested information in the Settings paragraph.

Point 9: Analysis section. I recommend that the authors include a Data Analysis section at the end of the Method to help the reader follow the results. This section should state the procedures and provide rationales for the authors’ analytic decisions.

Response 9: We added the requested section.

Point 10.  Table 2. I’d prefer to see the raw (i.e., un-collapsed) results.

Response 10: We appreciate your interest in the uncollapsed data. We originally thought we would present it with the 5 columns; however, in doing so, we felt that the patterns were more difficult to discern. Below we share the data for the reviewers’ benefit (see table), but ultimately, believe that the current tables present the most parsimonious picture.

Point 11: Chi-square results. Technically, chi-square results indicate proportional differences but not where the greater proportions are.

Response 11: We modified our statements to indicate what the Chi-square test reveals and what the pattern of data suggests. We also examined the standardized residuals to confirm which cells were contributors to the overall Chi-square value (i.e., those +/-2).

Point 12: Discussion. How do we know the extent to which teachers actually set an EQUITY goal?

Response 12: As described above, when setting a goal, teachers were required to select an equity-focused feature. Thus, equity goals are the only type of goal that could be set.

Point 13.  Additional data. Related to the previous point, is it possible to obtain counts of whether (and which) teachers clicked the learn more button to explore equity? I think this would be a really useful additional bit of data to include!

Response 13: This is a great idea. At this point in the development of the platform, we do not have a way to connect clicks on the Learn More pages to specific teachers or building users.

Point 14; Progress results. Include the n of those who never completed a goal review on p. 14).

Response 14: We added the requested information.

Point 15: Nesting. Were there any school-level differences in results that should be accounted for in the analyses?

Response 15: Given that our analyses were descriptive, we did not account for nesting within schools. We examined patterns across schools, however, we were hesitant to present the data and/or draw conclusions based on the small samples. Given the reviewers questions, we have added a comment indicating that those who set goals (33% came from School 1, 20% came from School 2, and 47% came from School 3), were proportional to the sample size in each school (34% participants were from School 1, 26% participants were from School 2, and 40% were from School 3).

In sum, I think the authors have done excellent work in developing this platform and applying it to equity in school discipline. I wish them all the best in continued study regarding their line of research.

Thank you.

Percent of General Education Teachers Who Report Various Responses for Frequency of Use of Each Strategy (N = 55)

Maximize Strategy

Rarely

Sometimes

Half of the Time

Often

Very Often

Total (n = 55)

Greetings

3.6%

3.6%

12.7%

36.4%

43.6%

Check-Ins

1.8%

9.1%

30.9%

43.6%

14.5%

Community Circles

3.6%

5.5%

12.7%

43.6%

34.5%

Classroom Expectations

0%

3.6%

5.5%

43.6%

47.3%

Acknowledge Pos. Behavior

0%

3.6%

14.5%

49.1%

32.7%

Corrective Feedback

0%

5.5%

41.8%

38.2%

14.5%

Teaching Prosocial

1.8%

7.3%

21.8%

56.4%

12.7%

Routines

0%

1.8%

5.5%

41.8%

50.9%

Effective Questioning

3.6%

3.6%

41.8%

41.8%

9.1%

Student Choice

1.8%

20%

30.9%

40%

7.3%

Reviewer 2

It has been a pleasure reading the article.

It has a concise abstract, correct introduction, clear materials and methods section, thorough results presentation, adequate discussion and appropriate conclusion which describes how to assess teachers’ use of equity-centered classroom management practices and how we can leverage interactive self-paced technology to move the needle on teachers’ implementation of equity-focused practices.

Response: Thank you.

Reviewer 2 Report

It has been a pleasure reading the article.

It has a concise abstract, correct introduction, clear materials and methods section, thorough results presentation, adequate discussion and appropriate conclusion which describes how to assess teachers’ use of 612 equity-centered classroom management practices and how we can leverage interac-613 tive self-paced technology to move the needle on teachers’ implementation of equity-fo-614 cused practices.

Author Response

Reviewer 2

It has been a pleasure reading the article.

It has a concise abstract, correct introduction, clear materials and methods section, thorough results presentation, adequate discussion and appropriate conclusion which describes how to assess teachers’ use of equity-centered classroom management practices and how we can leverage interactive self-paced technology to move the needle on teachers’ implementation of equity-focused practices.

Response: Thank you.

Round 2

Reviewer 1 Report

Thank you for the opportunity to review this manuscript again. I am impressed with the authors’ attention to detail, and they have addressed my main concerns adequately. My remaining suggestions are as follows:

1. Measure of teacher interest. On p. 8, line 343, the “and” is misplaced, making it look like a response option is missing.

2. Participating school characteristics. Can the authors provide data on PBIS fidelity of implementation for each school?

I congratulate the authors for their excellent work and look forward to further research on this approach.

Author Response

Thank you for the opportunity to review this manuscript again. I am impressed with the authors’ attention to detail, and they have addressed my main concerns adequately. My remaining suggestions are as follows:

Point 1: 1. Measure of teacher interest. On p. 8, line 343, the “and” is misplaced, making it look like a response option is missing.

Response 1: Thank you for noticing this. It has been fixed.

Point 2: Participating school characteristics. Can the authors provide data on PBIS fidelity of implementation for each school?

Response 2: Unfortunately, we did not collect a measure on PBIS fidelity from the school.